# Scope and Limitations of Exploiting the Ability of the Chemosensitizer NV716 to Enhance the Activity of Tetracycline Derivatives against *Pseudomonas aeruginosa*

**DOI:** 10.3390/molecules28114262

**Published:** 2023-05-23

**Authors:** Margot Draveny, Clémence Rose, Alexis Pinet, Laurent Ferrié, Bruno Figadère, Jean-Michel Brunel, Muriel Masi

**Affiliations:** 1MCT, INSERM U1261, UMR_MD1, Aix-Marseille Univ. & IRBA SSA, 27 Boulevard Jean Moulin, 13005 Marseille, France; margot.draveny@synchrotron-soleil.fr (M.D.); bruneljm@yahoo.fr (J.-M.B.); 2Synchrotron SOLEIL, L’Orme des Merisiers, Départementale 128, 91190 Saint-Aubin, France; 3BioCIS, Bâtiment H. Moissan, Université Paris-Saclay, CNRS, 91400 Orsay, France; clemence.rose@live.fr (C.R.); alexis.pinet9@gmail.com (A.P.); laurent.ferrie@universite-paris-saclay.fr (L.F.); bruno.figadere@universite-paris-saclay.fr (B.F.)

**Keywords:** Gram-negative bacteria, antibiotic adjuvant, NV716, antibiotic resistance, outer membrane

## Abstract

The spread of antibiotic resistance is an urgent threat to global health that requires new therapeutic approaches. Treatments for pathogenic Gram-negative bacteria are particularly challenging to identify due to the robust OM permeability barrier in these organisms. One strategy is to use antibiotic adjuvants, a class of drugs that have no significant antibacterial activity on their own but can act synergistically with certain antibiotics. Previous studies described the discovery and development of polyaminoisoprenyl molecules as antibiotic adjuvants with an OM effect. In particular, the compound NV716 has been shown to sensitize *Pseudomonas aeruginosa* to tetracycline antibiotics such as doxycycline. Here, we sought to explore the disruption of OM to sensitize *P. aeruginosa* to otherwise inactive antimicrobials using a series of tetracycline derivatives in the presence of NV716. We found that OM disruption expands the hydrophobicity threshold consistent with antibacterial activity to include hydrophobic molecules, thereby altering permeation rules in Gram-negative bacteria.

## 1. Introduction

There is a widely acknowledged need for new antibacterial agents to address the global increase in resistance, and this need is especially urgent to combat antibiotic-resistant Gram-negative bacteria [1,2]. In 2017, the World Health Organization (WHO) assigned the highest priority to antibacterial drug research and development against the Gram-negative bacteria *Acinetobacter*, *Pseudomonas,* and species of *Enterobacteriaceae* that are resistant to carbapenems and usually extensively drug-resistant (XDR) [3]. The same year, the WHO released a clinical pipeline report, which has been updated every year since then. Until 2021, 12 new antibacterial drugs have been approved. However, recently approved antibacterial agents show a limited degree of innovation as most of them are derivatives of existing classes and/or poorly address global public health priorities. The 2021 clinical antibacterial pipeline contains 77 antibiotics and/or combinations that include at least one new therapeutic entity. Of these, 45 are traditional antibacterial agents, including 27 against the WHO priority pathogens, and 32 are non-traditional [4].

The failure of the development of antibiotics active against Gram-negative bacteria is mainly due to the inability of small molecules to accumulate within these bacteria and reach a threshold concentration to inhibit their target [5,6]. Indeed, antibiotics are constrained to two opposite-direction fluxes across the Gram-negative bacterial envelope. First, Gram-negative bacteria are protected by an outer membrane (OM), which reduces the uptake of toxic compounds [7]. The OM is an asymmetric bilayer composed of lipopolysaccharides (LPS) in the outer leaflet and phospholipids in the inner leaflet. The tight packaging of LPS and its overall negative charge exclude large and hydrophobic molecules [8], and the influx of antibiotics is usually mediated by channel-forming proteins, called porins [7,9]. The general porins OmpF and OmpC of *Escherichia coli* and related enterobacteria are non-specific channels [10] and have quite large pores, allowing the passage of polar compounds below a size limit of ~600 Da, including many antibiotics [7,9]. However, other Gram-negative bacteria, such as *Pseudomonas aeruginosa* and *Acinetobacter baumannii*, do not have such large-channel porins and rely on substrate-specific channels to acquire small water-soluble compounds [7,11]. Consequently, their OM is approximately two orders of magnitude less permeable than that of *E. coli* [7]. In addition, antibiotics are also subject to an outgoing flux via transmembrane multidrug efflux pumps (i.e., AcrAB-TolC in *Enterobacteriaceae* and MexAB-OprM in *P. aeruginosa*) that expel them towards the medium [12]. As such, the synergy between limited OM diffusion and active efflux determines antibiotic accumulation and activity [13,14,15].

In recent years, antibiotic discovery efforts have attempted to increase the intracellular concentration of antibiotics inside Gram-negative bacteria through various approaches, including inhibition of efflux, disruption of the OM, and advances in medicinal chemistry. Regarding the latter aspect, recent studies have expanded the Gram-negative eNTRy “rules,” identifying rigid, flat molecules with a positive charge to be more compatible with porin-mediated uptake [16]. These concepts have been applied to modify antibiotics active against Gram-positive for Gram-negative activity [17,18,19]. While promising, this approach is limited to scaffolds that can be modified without loss of affinity for their target or modification of any other favorable pharmacological properties. An alternative approach is the use of adjuvants capable of disrupting OM integrity, thereby facilitating the entry and accumulation of many otherwise inactive antibiotics in Gram-negative bacteria [20]. Such compounds would immediately expand the range of treatments for Gram-negative pathogens [21]. 

Since their discovery in 1947, polymyxins have proven to attack the membranes of Gram-negative bacteria. Polymyxins were abandoned in the 1960s because of their nephrotoxicity but are now revived as the last-resort therapy for infections caused by XDR strains [22]. Quite recently, new polymyxin derivatives have been successfully studied preclinically for treating pneumonia caused by *P. aeruginosa* or *A. baumannii* in mouse models [23]. In 2017, the first OM disruptor, SPR741 (in-licensed by Spero Therapeutics and Northern Antibiotics), successfully passed the clinical phase 1 trial and was well tolerated at doses exceeding the dose that should be evaluated in the planned phase 2 [24,25].

Polyamines, like polymyxins, are cationic molecules that target Gram-negative membranes [26]. Among them, the polyaminoisoprenyl compound NV716 is a potent antibiotic adjuvant in several Gram-negative species. Previous studies showed that NV716 restores chloramphenicol and doxycycline activity against *P. aeruginosa* [27] and florfenicol activity against *Bordetella bronchiseptica* [28] in wild-type and clinical or farm strains. It also dramatically increases the activity of almost all the classes of antibiotics in *E. coli*, including that of the large, hydrophobic, and traditionally Gram-positive active antibiotics, such as rifampin and erythromycin; hydrophobic β-lactams such as oxacillin; chloramphenicol; tetracyclines; and nalidixic acid but not fluoroquinolones [Novelli, M.; Brunel, J.-M.; & Bolla, J.-M., in preparation]. NV716 has been shown to reduce active efflux of the 1,2′-DNA fluorescent probe in a dose-dependent real-time efflux assay [29]. However, NV716 is still active in efflux-deficient strains of *E. coli* and *P. aeruginosa*. In addition, several pieces of evidence indicate that NV716 binds to the LPS and acts by disrupting the OM integrity as a major barrier to antibiotic uptake [29,30,31] (Draveny, M.; Brunel, J.-M.; Jamme, F.; & Masi, M.; in preparation). One can assume this mode of action has much more impact on *P. aeruginosa* than on *E. coli*, whose OM contains the most effective porins.

In recent years, tetracyclines have been considered beyond their antibiotic activity [32]. In particular, the use of tetracyclines is associated with a reduced risk of developing Parkinson’s disease [33]. New tetracycline derivatives were recently synthesized with improved neuroprotective properties and reduced antibiotic activity [34,35]. Herein, we sought to determine whether the potential of OM disruption mediated by NV716 could be used as a therapeutic approach in combination with these new tetracycline derivatives. For this, we studied how the disturbance of the OM affects the entry rules of these compounds in *P. aeruginosa* and identified a significant expansion of the hydrophobicity threshold compatible activity.

## 2. Results

### 2.1. OM Perturbation Induced by NV716 Increases the Range of Hydrophobicity of Tetracycline Derivatives Compatible with Antibacterial Activity

NV716 was prepared as previously described [36] in 64% yield that includes a direct nucleophilic substitution of spermine on farnesyl chloride (Figure 1). 

Tetracyclines inhibit protein synthesis in Gram-positive and Gram-negative bacteria by preventing the attachment of aminoacyl-tRNA to the ribosomal acceptor (A) site [37]. This mechanism has been confirmed by X-ray crystallography [38]. Original tetracycline derivatives are shown in Figure 2. The rigid skeleton of tetracyclines contains four rings, a lower non-modifiable side that makes contact with the 30S sRNA, and an upper modifiable side (Appendix A). All tetracyclines that act as inhibitors of protein synthesis in bacteria require the amino group in position C4 (C4-dimethylamino group for optimal antibacterial activity) and keto-enolic tautomers in positions C1 and C3 of the A ring (R1 in Appendix A). COL-3, 4-dedimethylaminosancycline, or incyclinide (3), was the first chemically modified tetracycline that has been structurally rearranged to suppress antibacterial properties by eliminating the C4-dimethylamino group in the A ring while preserving other activities of interest [39]. The D ring is the most permissive for changes. Thus, it can be hypothesized that all subsequent modifications could affect the bacterial activity due to the modification of clogD (R2-R5 in Appendix A). 

We screened 28 tetracycline derivatives, including three known tetracycline antibiotics and 25 chemically modified derivatives, for their antibacterial activity in *P. aeruginosa* PAO1 and their degree of potentiation by the OM disruptor NV716. In the absence of NV716, the three known tetracycline antibiotics (**3a**–**3c**) showed modest activity with MICs ranging from 6.25 to 12.5 µg/mL, while all the chemically modified derivatives (**3d**–**3ab**) were completely inactive with MICs up to 200–400 µg/mL or higher (Figure 3A and Appendix A). Of the 28 compounds tested, 18 were potentiated by NV716 with a reduction in MIC of more than 5-fold in bacteria treated with NV716 compared to untreated bacteria (Figure 3A and Appendix A). As shown previously, NV716 at a concentration of 10 µM consistently decreased the MIC of doxycycline (**3c**) by 62-fold in *P. aeruginosa* PAO1 [27,30]. We next determined the lipophilicity (cLogD, calculated partition coefficient cLogD at pH 7.4) for all the tested compounds and ranked them according to their ability to be potentiated by the addition of NV716 (Figure 3B). As mentioned above, the loss of antibacterial activity of COL-3 and the other derivatives is likely attributable to the removal of the C4-dimethylamino group in the A ring. This modification also increases the hydrophobicity of most of the compounds. Nevertheless, the addition of NV716 expanded the range of cLogDs compatible with antibacterial activity towards more hydrophobic compounds (Figure 3B). Indeed, the MIC values of tetracycline derivatives with a cLogD value between −1.8 and −4.3 are either reduced to a level below the sensitivity threshold of *P. aeruginosa* (MIC < 2 μg/mL for compounds **3a**–**3c**; Figure 3B, blue dots) or potentiated from 16 to 128-fold (compounds 3d–3n, Figure 3B, green dots). On the other hand, tetracycline derivatives with a cLogD value above −1.8 cannot be potentiated (compounds **3s**–**3ab**, Figure 3B, red dots). Some compounds with a cLogD very close to −1.8 are less well potentiated, from 4 to 32-fold, in the presence of NV716 (compounds **3o**–**3r**, Figure 3B, yellow dots). 

### 2.2. Antibacterial Activity of the Tetracycline Derivatives correlates with Increased Uptake in the Presence of NV716

The OM is a main barrier to the diffusion of antibiotics. Therefore, we next examined how OM perturbation by NV716 affects the accumulation of tetracycline derivatives inside bacteria. For this, we chose doxycycline (**3c**) as a positive control, and **3j** and **3u** as representative compounds respectively potentiated or not by NV716. It is worth noting that the chemical structure **3u** is not fully represented (Figure 2) since it is patent pending in another context. This compound was chosen for this study because of its fluorescence properties and lack of antibacterial activity (i.e., cLogD is incompatible with potentiation by NV716). The fluorescence of doxycycline increases when it binds to its target (i.e., the ribosomes) inside the bacteria. We recently set up a doxycycline uptake assay with intact *P. aeruginosa* cells and demonstrated that NV716 causes an increase in doxycycline uptake (as shown by an increase in fluorescence, AUC × 3.6), indicating that NV716 helps doxycycline get inside the bacteria [Draveny, M.; et al., in prep.] (Figure 4A). Here, NV716 also increased the uptake of **3j** (AUC × 4.2) but not **3u**, consistent with the ability of NV716 to potentiate or not their antibacterial activity (Figure 4B). Together, these results show that disruption of OM by NV716 overcomes the intrinsic resistance of *P. aeruginosa* to hydrophobic compounds as it helps to increase their accumulation and activity. However, these observations could also result from a difference in the affinity of the molecules for their target. Therefore, we elucidated the inhibition of translation in an *E. coli* cell-free transcription-translation system using GFP as a reporter protein. Here, we found that compounds **3j** and **3u** similarly inhibit the production of GFP (inhibitory concentration 50%, IC_50_ = 575.2 ± 1.07 µM and IC_50_ = 603.3 ± 1.39 µM, respectively) (Figure 5). Noteworthy, doxycycline was approximately 30-fold more active in inhibiting bacterial translation compared to the tetracycline derivatives (IC_50_ = 19.71 ± 1.38 µM) (Figure 5). 

## 3. Materials and Methods

### 3.1. Bacterial Strains and Culture Conditions

*P. aeruginosa* strains PAO1 and PAS263 (PAO1Δ*pvdF*) [40] were used for the experiments. The bacterial cultures were carried out routinely in LB-Miller medium or Mueller-Hinton 2 (MH2) at 37 °C with shaking (160 rpm).

### 3.2. Reagents

Chlortetracycline (**3a**), demeclotetracycline (**3b**), and doxycycline (**3c**) were purchased from Merck. COL-3 (**3j**) (also named CMT-3, 4-dedimethyl aminosancycline, or incyclinide) was purchased from MedChemExpress (Monmouth Junction, NJ, USA). 9-Tert-butyl doxycycline (**3x**) was ordered from Echelon Biosciences Research Labs. DDMC (**3o**), RDOX (**3l**), DDOX (**3k**), and NV716 were synthesized as described previously [34,35,36]. The procedures for compounds **3g**–**3i**, **3l**–**3n**, **3q**–**3w**, and **3y**–**3ab** were described previously [Rose, C.; Ferrié, L.; Tomas-Grau, R. H.; Zabala, B.; Brunel, J.-M.; Michel, P. P.; Chehin, R.; Raisman-Vozari, R.; Figadère, B. Design and Synthesis of New Non-Antibiotic Doxycyclin Derivatives Against Alpha-Synuclein Aggregation with Anti-Inflammatory Properties. ChemRxiv. https://doi.org/10.26434/chemrxiv-2023-n8prz, accessed on 25 April 2023]. Procedures and characterization data for the synthesis of compounds RDMC (**3i**), DDMC (**3o**), RT (**3e**), DCT (**3p**), and RCT (**3d**) are described below.

Synthesis of RDMC (**3i**) and DDMC (**3o**): In a 50 mL round-bottom flask, demeclocycline hydrochloride (500 mg, 1.0 mmol, 1.0 equiv) was suspended in AcOH (50% aqueous) (50 mL), then zinc (powder) (706 mg, 10.0 mmol, 10 equiv) was added, and the reaction mixture was stirred at room temperature for 2 h. The resulting solution was filtered through a small pad of Celite with AcOH. The organic phase was extracted with CH_2_Cl_2_, washed with HCl (1 M) and brine, dried over MgSO_4_, filtered off, and concentrated in vacuo. The crude mass was 268 mg. Purification with combiflash chromatography [60 g SiO_2_, elution 0.5% (acetone + 1% formic acid) in CH_2_Cl_2_ to 10% over 50 min] afforded DDMC (**3o**) (32 mg) followed by RDMC (**3i**) (97.5 mg). RDMC:
^1^H NMR (400 MHz, Acetone-*d_6_*) δ 18.47 (s, 1H, C_3_-OH), 14.91 (s, 1H, C_12_-OH), 11.89 (s, 1H, C_10_-OH), 9.10 (brs, 1H, NH_2_), 7.63 (brs, 1H, NH_2_), 7.58 (d, *J* = 9.0 Hz, 1H, H8), 6.95 (d, *J* = 9.0 Hz, 1H, H9), 5.62 (s, 1H, C_12a_-OH), 4.95 (dd, *J* = 5.6, 2.8 Hz, 1H, H_6_), 4.52 (d, *J* = 6.0 Hz, 1H, C_6_-OH), 3.23 (dd, *J* = 19.4, 6.4, 1H, H_4_), 3.06 (ddd, *J* = 9.8, 6.3, 2.9 Hz, 1H, H_5a_), 2.65–2.54 (m, 2H, H_4_+H_4a_), 2.25–2.06 (m, 2H, H_5_). ^13^C NMR (101 MHz, Acetone) δ 196.09, 193.94, 193.17, 179.02, 175.27, 162.28, 142.05, 138.00, 123.83, 120.03, 117.11, 105.52, 99.12, 76.45, 66.05, 38.46, 36.95, 35.76, 27.42. HRMS (ESI): calculated for C_19_H_16_ClNO_8_ [M+H]^+^: 422.0637, found 422.0646. 

Synthesis of RT (**3e**): In a 1 L round-bottom flask, tetracycline (10 g, 22.5 mmol, 1.0 equiv) was suspended in AcOH (50% aqueous) (400 mL), then HCl (2.2 mL, 27.0 mmol, 1.2 equiv, 37%) was added, followed by the addition of zinc powder (14.8 g, 225.1 mmol, 10 equiv), and the reaction mixture was stirred at room temperature for 3 h. The resulting solution was filtered through a small pad of Celite with AcOH. The organic phase was extracted with CH_2_Cl_2_, washed with HCl (1 M) and brine, dried over MgSO_4_, filtered off, and concentrated in vacuo. Purification with combiflash chromatography [300 g SiO_2_, elution 0.5% (acetone + 1% formic acid) in CH_2_Cl_2_ to 10% over 50 min] afforded RT (**3e**) (3.71 g). ^1^H NMR (300 MHz, DMSO-*d_6_*) δ 15.31 (s, 1H, C_12_-OH), 11.88 (s, 1H, C10-OH), 8.97 (brs, 1H, NH_2_), 8.70 (brs, 1H, NH_2_), 7.53 (t, *J* = 8.0 Hz, 1H, H_8_), 7.09 (d, *J* = 7.7 Hz, 1H, H_9_), 6.91 (d, *J* = 8.3 Hz, 1H, H_7_), 6.58 (brs, 1H, C_12a_-OH), 4.86 (s, 1H, C_6_-OH), 3.17 (brd, *J* = 17.9 Hz, 1H, H_4_), 2.77 (dd, *J* = 11.0, 5.4 Hz, 1H, H_5a_), 2.47–2.28 (m, 2H, H_4_+H_4a_), 1.97 (m, 1H, H_5_), 1.79 (q, *J* = 12.6 Hz, 1H, H_5_), 1.49 (s, 3H, C_6_-Me). ^13^C NMR (75 MHz, DMSO) δ 194.85, 192.87, 191.76, 178.11, 173.60, 161.40, 148.08, 136.30, 116.91, 115.22, 114.57, 106.36, 97.4, 74.78, 67.99, 41.88, 35.35, 34.71, 24.89, 22.52. HRMS (ESI): calculated for C_20_H_18_NO_7_ [M+H−H_2_O]^+^: 384.1078, found 384.1082. 

Synthesis of DCT (**3p**) and RCT (**3d**): In a 50 mL round-bottom flask, chlortetracycline hydrochloride (500 mg, 0.97 mmol, 1.0 equiv) was suspended in AcOH (10 mL) followed by water (roughly 1 mL) until the dissolution of the product. Then, zinc (powder) (635.4 mg, 9.7 mmol, 10 equiv) was added, and the reaction mixture was stirred at room temperature for 2 h. The resulting solution was filtered through a small pad of Celite with AcOH. The organic phase was extracted with CH_2_Cl_2_, washed with HCl (1 M) and brine, dried over MgSO4, filtered off, and concentrated in vacuo. The crude mass was 275 mg. Purification with combiflash chromatography [60 g SiO_2_, elution 0.5% (acetone + 1% formic acid) in CH_2_Cl_2_ to 10% over 50 min] afforded DCT (**3p**) (35 mg) followed by RCT (**3d**) (165 mg). DCT (**3p**): ^1^H NMR (300 MHz, Acetone-*d_6_*) δ 18.50 (d, *J* = 1.1 Hz, 1H), 14.80 (s, 1H), 12.75 (s, 1H), 9.26 (s, 1H), 7.66 (s, 1H), 7.60 (d, *J* = 9.0 Hz, 1H), 6.92 (d, *J* = 9.0 Hz, 1H), 4.68 (d, *J* = 1.0 Hz, 1H), 3.94–3.82 (dd, *J* = 4.4, 2.0 Hz, 1H), 2.87–2.72 (m, 2H), 2.55 (m, 1H), 2.51–2.44 (m, 1H), 2.39 (dd, *J* = 16.6, 13.7 Hz, 1H), 1.95 (d, *J* = 1.6 Hz, 3H), 1.06 (td, *J* = 12.7, 11.2 Hz, 1H). ^13^C NMR (75 MHz, Acetone-*d_6_*) δ 201.89, 197.55, 191.35, 174.85, 169.11, 163.13, 144.43, 141.34, 123.06, 119.35, 116.38, 105.88, 100.01, 72.15, 51.89, 47.79, 39.96, 31.70, 28.59, 27.78. HRMS (ESI): calculated for C_20_H_19_ClNO_7_ [M+H]^+^: 420.0845, found 420.0850. RCT (**3d**): ^1^H NMR (400 MHz, DMSO-*d_6_*): δ 18.41, (s, 1H, C_3_-OH), 15.20 (s, 1H, C_12_-OH), 12.24 (s, 1H, C_10_-OH), 8.95 (s, 1H, NH_2_), 8.72 (s, 1H, NH_2_), 7.52 (d, 1H, *J* = 9.0 Hz, H_8_), 6.93 (d, *J* = 9.0 Hz, 1H, H_7_), 6.63 (s, 1H, C_12a_-OH), 5.26 (s, 1H, C_6_-OH), 3.16 (brd, 1H, *J* = 15.9 Hz, H_4_), 2.86 (dd, *J* = 11.0, 5.4 Hz, 1H,H_5a_), 2.35-2.47 (m, 2H, H_4_+H_4a_), 2.01 (brd, *J* = 10.8 Hz, 1H, H_5_), 1.81 (s, 3H, C_6_-Me), 1.70-1.84 (m, 1H, H_5_) ppm. ^13^C NMR (100 MHz, DMSO-_d6_): δ 194.85, 191.67, 178.84, 173.53, 160.72, 143.72, 139.45, 121.31, 118.71, 118.10, 117.07, 105.39, 97.34, 74.77, 70.43, 42.28, 35.53, 34.75, 25.01, 24.84. HRMS (ESI): calculated for C_20_H_19_ClNO_8_ [M−H_2_O+H]^+^: 418.0688, found 436.0681. 

### 3.3. Potentiation Assays

MICs were conducted in PAO1 in at least three biological replicates following the CLSI protocol. The fold reduction of MIC was determined by dividing the MIC of the antibiotic alone by its MIC in the presence of 10 µM NV716. 

### 3.4. Physicochemical Property Calculations

Structure analysis was conducted and clogD at pH 7.4 was calculated using Marvin Suite, ChemAxon (Budapest, Hungary).

### 3.5. Accumulation Assays

PAS263 is an isogenic mutant of PAO1 that carries a chromosomal deletion of *pvdF,* which encodes the pyoverdine siderophore. Accumulation assays were performed in PAS263 instead of PAO1, as pyoverdine’s spectral properties overlap with those of tetracycline derivatives. Here, bacteria were grown to an exponential phase. The cultures were centrifuged at 6000× *g* for 20 min at 20 °C and the pellets were resuspended in NaPi-MgCl_2_ buffer (50 mM sodium phosphate pH 7.4; 2 mM MgCl_2_) to obtain 0.6 UDO/mL. Tetracycline derivatives and NV716 were used at final concentrations of 100 µM and 10 µM, respectively. For these assays, the order of the addition of the reagents is important to limit the loss of fluorescence recording at time zero. Thus, 10 µL of individual 10X tetracycline derivative solutions, 10 µL of NaPi-MgCl_2_ buffer or 10X NV716, and 180 μL of bacterial suspensions were added sequentially to the wells of a 96-well microplate with black sides and a clear bottom (Thermo Fisher Scientific, ref. 165305, Illkirch-Graffenstaden, France). An additional well contains bacteria alone. The fluorescence (A. U.) of the tetracycline derivatives was immediately measured every minute for 30 min with a TECAN Spark microplate reader using specific wavelengths for excitation and emission as determined in Appendix A. The obtained data were then processed to obtain A. U. = f(t). The fluorescence of the cells alone measured at each time was subtracted for all the conditions tested during the same experiment. Finally, the fluorescence at time zero was subtracted for each time point of each condition. The uptake of tetracycline derivatives was quantified as the area under the curve (AUC) by using GraphPad Prism 6 software. Independent experiments were performed at least three times, and the results are shown as the means of the replicates.

### 3.6. Bacterial Cell-Free Expression Assays

The effect of the selected tetracycline derivatives (i.e., **3c**, **3j**, and **3u**) on in vitro bacterial protein synthesis was tested using the Expressway Cell-Free *E. coli* Expression System (Invitrogen for Thermo Fisher Scientific, Illkirch-Graffenstaden, France). pEXP5-CT TOPO-GFP was used as a template for in vitro transcription–translation. The reactions were performed following the manufacturer’s instructions and carried out in 50 µL including the addition of feed buffer at 30 min of incubation to support optimal protein synthesis. Reactions were initiated with 1 µg of plasmid DNA; mixtures were incubated at 30 °C with shaking (300 rpm), and fluorescence was measured after 3 h with a TECAN Spark microplate reader (λex = 475 nm; λem = 520 nm). IC_50_ values were calculated using GraphPad Prism 6 software. Independent experiments were performed at least three times, and the results are shown as the means of the replicates ± SD.

## 4. Conclusions

New antibacterial discoveries and developments have not kept pace with the spread of resistance. In particular, the membrane impermeability of Gram-negative bacteria represents a major challenge to overcoming the penetration and activity of both existing and new drugs. Perturbation of the OM barrier by chemical or genetic disruption has been shown to increase the sensitivity of Gram-negative bacteria to many antibiotics that are traditionally active against Gram-positive bacteria. Indeed, several groups have published proof-of-concept studies using cationic peptides [23,41], small molecules [26,42], LPS chelators [43], and genetic manipulations [13,44] that support this approach. Despite an upsurge in efforts in this area, as evidenced by reports on the global preclinical antibacterial pipeline, previous studies have mainly focused on studying the effectiveness and characterization of the mode of action of potentiating molecules taken individually [45]. However, this research field lacks studies of the strengths and limits of this approach. In 2020, Eric Brown and colleagues published a comprehensive study to analyze the interaction between OM perturbations and general constraints for effective antibiotic treatment. Interestingly, OM disruption was shown to overcome intrinsic, acquired, and spontaneous antibiotic resistance in *E. coli*, thus validating this approach [46]. 

The polyaminoisoprenyl compound NV716 was shown to revive old, disused antibiotics such as doxycycline against *P. aeruginosa* [27,30]. In this work, we focused on the impact of OM perturbation caused by NV716 to potentiate the activity of a new series of original tetracycline derivatives in this species. We found that OM disruption increased susceptibility towards hydrophobic compounds; this potentiation of activity was restricted to a specific range of cLogD and reflected an increase in uptake without affecting the affinity to the target. 

Other mechanisms, including antibiotic inactivation, target modification, and/or overexpression of multidrug efflux pumps, are also responsible for acquired resistance in Gram-negative pathogens. Thus, we will next seek to study the efficacy of NV716 as an OM disruptor against clinical isolates of *P. aeruginosa*.

## Figures and Tables

**Figure 1 molecules-28-04262-f001:**
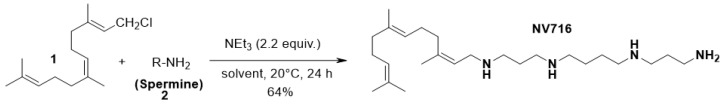
Synthetic pathway of the preparation of polyaminoisoprenyl compound NV716.

**Figure 2 molecules-28-04262-f002:**
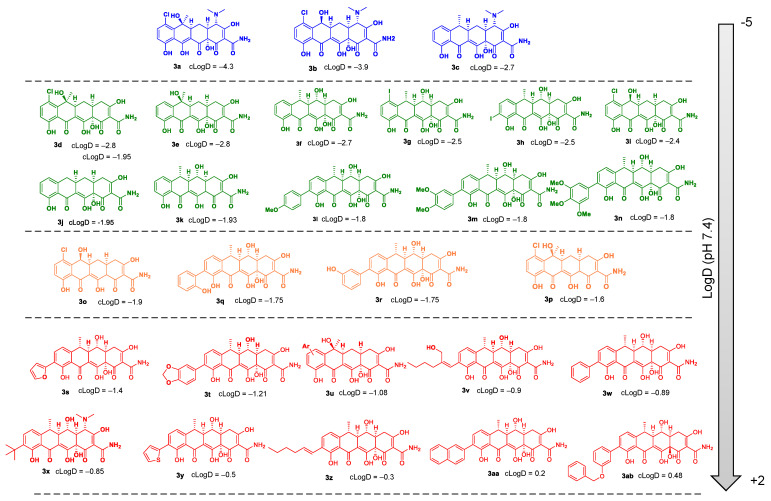
Chemical structures and cLogD values of original tetracycline derivatives **3a**–**3ab**.

**Figure 3 molecules-28-04262-f003:**
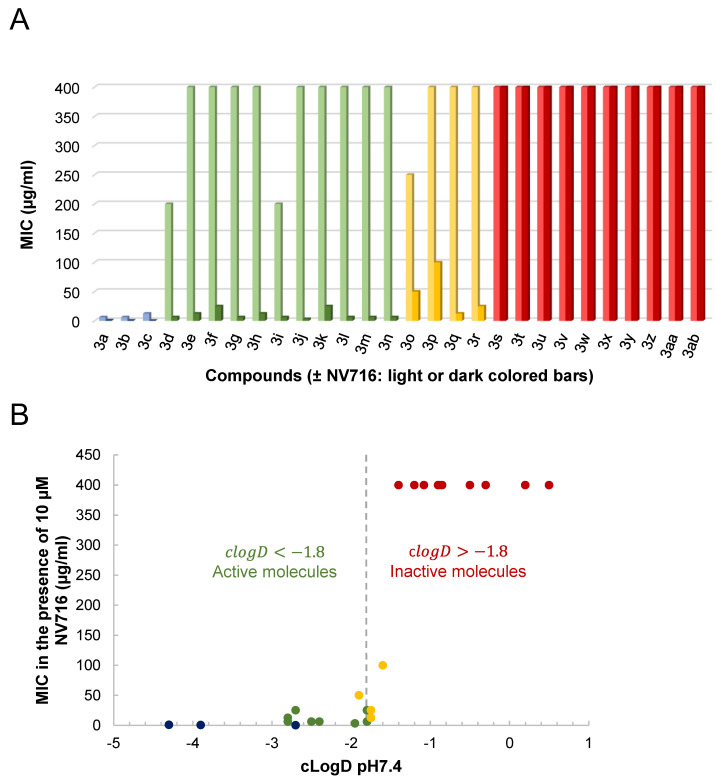
Changes in drug activity following OM disruption by NV716. (**A**) shows that tetracycline derivatives can be potentiated (reduction in MIC ≥ 5 to 128-fold) or unaffected (reduction in MIC < 5-fold) by NV716 added at a final concentration of 10 µM. (**B**) shows the relationship between the MIC determined in the presence of NV716 and the cLogD at pH 7.4. Compounds are colored as follows: only existing tetracycline antibiotics show some antibacterial activity on *P. aeruginosa* in the absence of NV716 (**3a**, **3b,** and **3c** in blue); most of the new tetracycline derivatives are either inactive (red) or active (green) in the presence of NV716 on either side of a virtual cLogD value of −1.8; a few exceptions are weakly active with cLogD~−1.8 (yellow).

**Figure 4 molecules-28-04262-f004:**
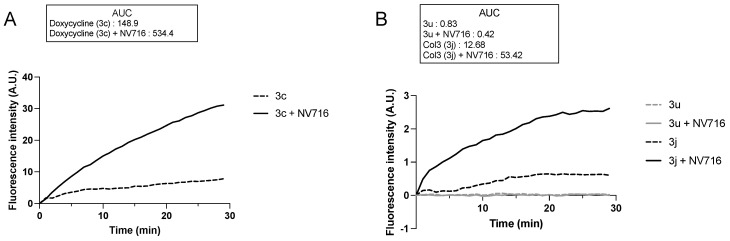
OM disruption by NV716 increases the uptake of some but not all tetracycline derivatives. The concentration of the tested tetracycline derivatives was 100 µM, which was not inhibitory for an OD_600_ ~ 0.6 during the experiment. The concentration of NV716 was added as indicated in Materials and Methods. Increased doxycycline uptake in *P. aeruginosa* PAO1 in the presence of NV716 has been reported and used as a positive control. Due to the difference in the y-axis scale, the uptake of **3c** is shown in (**A**); **3u** and **3j** are shown in (**B**). Bacterial cells without treatment did not show any increase in fluorescence, indicating the absence of intrinsic fluorescence. Fluorescence was monitored as described in Materials and Methods. Data are normalized concerning the fluorescence of individual tetracycline derivatives at time zero. Data are represented as the mean of at least three independent experiments. A. U., arbitrary units.

**Figure 5 molecules-28-04262-f005:**
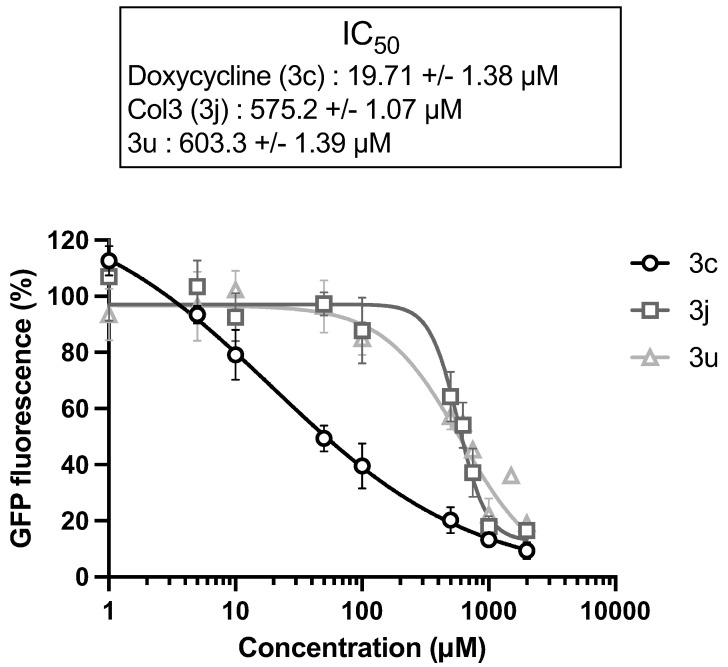
Tetracycline derivatives with different physicochemical properties interfere with protein synthesis in vitro. Inhibition of the synthesis of GFP as a reporter protein was monitored in an *E. coli* cell-free transcription-translation system. Doxycycline is a known inhibitor of protein synthesis and was used as a positive control. Data are represented as the mean of at least three independent experiments, and error bars represent the SD of replicates.

## Data Availability

The data presented in this study are available on request from the corresponding author.

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
