# Peer review of "Scope and Limitations of Exploiting the Ability of the Chemosensitizer NV716 to Enhance the Activity of Tetracycline Derivatives against Pseudomonas aeruginosa"

_molecules, 2023, doi:10.3390/molecules28114262_

Round 1

Reviewer 1 Report

The paper “Scope and limitations of exploiting the ability of the chemosensitizer NV716 to enhance the activity of tetracycline derivatives against Pseudomonas aeruginosa” shows how the addition of a compound NV716 can decrease the MIC of certain tetracycline derivatives and also enhance their activity by OM perturbation.

Overall, the paper describes the method and appropriate results adequately. However, I would like to make one suggestion here. Since the introduction talks about antibacterial resistance, it is important to see the effect on biofilm. Although it maybe tedious to screen all 28 compounds, but it would add value to the study if the effect on biofilm is studied in at least positive compounds.

What are the controls in the MIC testing study?

Author Response

We thank the reviewer for his/her comments and suggestions. Several studies have already been published to characterize the mode of action of NV716 on P. aeruginosa and others are in preparation. In particular, two original research articles with our collaborator Françoise Van Bambeke (Université Catholique de Louvain, Belgium) have reported that NV716 enhances antibiotic activity against biofilms of several Gram-negative species, including P. aeruginosa. Ciprofloxacin and rifampicin but not doxycycline have been tested in these assays (Wang G, Brunel JM, Preusse M, Mozaheb N, Willger SD, Larrouy-Maumus G, Baatsen P, Häussler S, Bolla JM, Van Bambeke F. The membrane-active polyaminoisoprenyl compound NV716 re-sensitizes Pseudomonas aeruginosa to antibiotics and reduces bacterial virulence. Commun Biol. 2022, 5:871. doi: 10.1038/s42003-022-03836-5; Wang G, Brunel JM, Rodriguez-Villalobos H, Bolla JM, Van Bambeke F. The polyaminoisoprenyl potentiator NV716 revives disused antibiotics against Gram-negative bacteria in broth, infected monocytes, or biofilms, by disturbing the barrier effect of their outer membrane. Eur J Med Chem. 2022, 238:114496. doi: 10.1016/j.ejmech.2022.114496).

As previously observed, NV716 significantly reduced the MICs of many antibiotics, with the highest synergy observed with the most lipophilic drugs. We wanted to explore this feature in more detail. Consequently, our work aimed to establish a relationship between the capacity of NV716 to potentiate the activity of antibiotic molecules and the hydrophobic nature of these same molecules, all belonging to the same family. This was done on planktonic bacteria.

First, we agree with the reviewer that biofilm formation is an important trait of P. aeruginosa’s lifestyle and pathogenicity. As suggested, we could analyze biofilm formation when bacteria are exposed to doxycycline with or without NV716 (biofilm biomass using crystal violet staining). However, this will need more than 10 days to perform triplicates of these experiments.

Second, the determination of MICs is carried out routinely in our laboratory. Control experiments always include bacteria only (growth control) and medium only (medium sterility control).

Reviewer 2 Report

The manuscript "Scope and limitations of exploiting the ability of the chemosensitizer NV716 to enhance the activity of tetracycline derivatives against Pseudomonas aeruginosa" by Margot Draveny et al. presents the effect of a synthetic compound, NV716, on the activity of 28 tetracyclines derivatives against P. aeruginosa. The authors demonstrate that treatment with 10 μΜ NV716 decreased the MIC of the otherwise inactive antimicrobials up to 128-fold, including a 10-60-fold reduction in the MIC of three pharmaceutical tetracylines. In addition, the authors demonstrate an increased doxycycline uptake in P. aeruginosa upon addition of NV716, using a fluorescence uptake assay of intact cells.

Overall the findings of this work are very interesting, albeit the project lacks any novelty, given that the discovery, synthesis and effect of NV716 have been described elsewhere. Still, the work is carried out appropriately and is presented in a decent way, which could be improved further. I reckon that the results of this work could be of interest to the readers of Molecules, however, the authors should address the following issues in order to improve their manuscript and meet the quality standards of Molecules.

1. Given that the tetracyclines used are numbered as compounds 3a-3ab (Figure 2), their label should be in bold throughout the manuscript and Tables and Figures.

2. LogD should be replaced by cLogD throughout the manuscript, wherever required, and also in Figure 2. 

3. Fonts in Figure 2 are too small and should be increased. Also, the right arrow's labels "-5" and "+2" are not useful, instead dashed lines could be indicated at -2.7, -1.8, -1.75 and 0.5.

4. Error bars are missing from Figure 3 panel A, and standard deviation should be given in Supporting Table S1.

5. In Figure 3, panel B is not really useful. Instead the authors could present a color-coded bar chart showing the fold decrease of MIC upon treatment with NV716. Color-coding of the bars will clearly indicate the cLogD threshold.

6. Molecular van der Waals volumes of the tetracyclines have been calculated and presented in Table S1, but there's no mention or discussion about their correlation with MIC with or without NV716.

7. Given that the fluorescence signal of 3u is negligible (Figure 4B) and that it's structure is not even disclosed, why the authors didn't try any other similar derivative (MIC > 200, similar cLogD and volumes), such as 3s, 3t, 3w, 3y, 3aa, 3ab. All these compounds contain an aryl ring on ring D of the tetracycline core.

8. In general, I suggest against the publication of compounds that are not disclosed due to patents pending. In particular for this case, 3u does not add any data other than those provided by compounds 3s-3ab.

9. The last statement of the abstract: "the outer membrane disruption can sort molecules according to their hydrophobicity, thus changing the rules of drug permeation compatible with antibacterial activity" is rather superficial and inaccurate. NV716 disrupts the OM and allows higher uptake of tetracyclines with cLogD < 1.8. It does not change the rules of drug permeation, neither does it sort the compounds according to their lipophilicity. Please revise accordingly.

10. Lines 102-105 of the last paragraph in the Introduction should be elaborated more with regard to the other effects of tetracyclines mentioned in a separate paragraph. Then lines 106-110 should be a separate paragraph, which could contain more information.

Apart from one of the last points (9) mentioned above, there are numerous grammatical issues with the manuscript. I will list only a handful, however, revision of the manuscript by a native English speaker will significantly improve its quality.

Line 15: "...makes exploring avenues that breathe new life into existing drugs imperative" is too poetic for an abstract.

Line 21: "-NV referring to Nouvelle Vie/New Life-" is a rather useless detail for an abstract.

lines 123-124: "All tetracyclines that act as inhibitors of protein synthesis in bacteria need the amino group in position C4", need is not used appropriately.

lines 129-130: "The D ring is the most flexible to changes", flexible is not used appropriately here.

lines 291-292: van der Walls? or Waals.

Author Response

See the corresponding attached file.

Round 2

Reviewer 2 Report

The manuscript has been significantly improved after the authors revised most of the issues I had suggested. English language is fine now, and the overal quality of the manuscript meets the standards of Molecules.